# Gamma-Aminobutyric Acid Signaling in Damage Response, Metabolism, and Disease

**DOI:** 10.3390/ijms24054584

**Published:** 2023-02-26

**Authors:** Kimyeong Kim, Haejin Yoon

**Affiliations:** Department of Biological Sciences, Ulsan National Institute of Science and Technology, Ulsan 44919, Republic of Korea

**Keywords:** gamma-aminobutyric acid, metabolite, neurotransmitter, liver disease, cancer

## Abstract

Gamma-aminobutyric acid (GABA) plays a crucial role in signal transduction and can function as a neurotransmitter. Although many studies have been conducted on GABA in brain biology, the cellular function and physiological relevance of GABA in other metabolic organs remain unclear. Here, we will discuss recent advances in understanding GABA metabolism with a focus on its biosynthesis and cellular functions in other organs. The mechanisms of GABA in liver biology and disease have revealed new ways to link the biosynthesis of GABA to its cellular function. By reviewing what is known about the distinct effects of GABA and GABA-mediated metabolites in physiological pathways, we provide a framework for understanding newly identified targets regulating the damage response, with implications for ameliorating metabolic diseases. With this review, we suggest that further research is necessary to develop GABA’s beneficial and toxic effects on metabolic disease progression.

## 1. Introduction

Metabolites are mainly used as energy sources and cellular building blocks. Sometimes, they are directly involved in signaling pathways. Gamma-aminobutyric acid (GABA) is a key signaling molecule and neurotransmitter in the brain system [1]. Since GABA affects neuronal activity, GABA-transaminase (GABA-T) inhibitors and GABA agonists have been used to treat brain diseases. GABA levels have been linked to metabolic organs and the progression of metabolic diseases. GABA plays important roles not only in the brain but also in different metabolic organs. We will focus on its roles in metabolic organs to understand the mechanisms associated with GABA signaling and dysfunction in metabolic diseases caused by excessive lipid accumulation. The harmful effects of lipid overload can lead to replication stress, consequently causing DNA damage in the liver [2], which is directly regulated by GABA metabolism. Cellular levels of GABA regulate ion-dependent transporters in the liver [3]. GABA receptors (GABARs) co-ordinate hepatocyte depolarization and hyperpolarization with lipid accumulation in metabolic disorders, including hyperinsulinemia and insulin resistance. The toxic effect of hepatic GABA can improve insulin resistance by inhibiting skeletal muscle glucose clearance. GABA released from island β cells can act on α cells and β cells through the paracrine and autocrine pathways. GABA stimulation can affect β-cell membrane depolarization and activate the PI3K/Akt signaling pathways [4], thus playing an important role in cellular growth and differentiation. It is important to understand the hepatic GABA function and the action of GABARs in various disease conditions. GABA is generated from glutamate through glutamate decarboxylase (GAD) enzymes, which are expressed differently in various organs, leading to the following new questions: (1) How does GABA have beneficial or toxic effects in different metabolic organs? (2) What is the upstream regulator that modulates GABA function? and (3) How do intracellular GABA intermediates directly contribute to cellular dysfunction? In this review, we will highlight the emerging role of GABA in various metabolic organs and its beneficial and toxic effects on disease progression.

## 2. Cellular Function of GABA in the Brain

Among metabolites, GABA is known as a signaling molecule. GABA is an inhibitory neurotransmitter by regulating ionic channel inflow and outflow. Both glutamate and GABA are key neurotransmitters in the brain, including the overall level of excitement in the brain. The levels of these metabolites are important for maintaining physiological homeostasis. Since long-term imbalances in these metabolites can lead to brain diseases [5], an understanding of the regulation of GABA uptake is necessary. The GABA-A receptor is a quintessential ligand-gated ion channel that mediates synaptic inhibition throughout the central nervous system [6]. After GABA performs its action as a neurotransmitter, it is reabsorbed into the presynaptic neurons and neural bridges [7].

Since a GABA-T inhibitor causes GABA accumulation in the brain by increasing extracellular GABA concentrations and inhibiting neuronal activity [8], many research studies have been conducted to investigate the control of GABA and glutamate in neurons. GABAergic, a neuron which produces GABA, and glutamate neurons play an important role in the activity of gonadotropin-releasing hormone (GnRH) neurons by maintaining stable rates of glutamate receptor synaptic transfer to ionic GABA and GnRH neurons in various estrogen feedback situations [9]. Glutamate and GABA were shown to depolarize cells in the ventricular region of the mouse embryo neocortex in the early stages of cortical neurogenesis. Glutamate can selectively interact with α-amino-3-hydroxy-5-methyl-4-isoxazole propionic acid (AMPA) and kainate-type glutamate receptors (AMPARs and KARs) and bind to GABA-B receptors. GABA and glutamate can increase the concentration of Ca^2+^ in ventricular zone cells but decrease DNA synthesis by activating voltage-gated Ca^2+^ channels. GABA and glutamate can boost DNA synthesis during treatment with GABA-A and AMPA/Kainate receptor antagonists [10] (Figure 1). Therefore, it is important to understand how GABA regulates the levels of ions with receptors and the mechanism of signal transmission.

Three types of GABARs have been identified: A, B, and C. Types A and C are ligand Cl^−^ inflow channels. Type B activates potassium outflow channels through G-protein signaling to induce hyperpolarization. Tension inhibition mediated by the GABA-A receptor can decisively regulate neuronal excitability and brain function [11]. GABA-A is required to trigger the Ca^2+^ response. For example, GABA was shown to activate ionic GABARs to stimulate cerebrovascular angiogenesis and promote neurovascular binding in cerebrovascular endothelial cells [12]. The mechanism of GABA is that the GABARs-dependent Ca^2+^ response leads to promoting cerebrovascular angiogenesis and inducing neurovascular binding for activating brain function. Signaling cascades consisting of GABA-B receptor G protein and G-protein-gated K+ channel (GIRK), which induce hyperpolarization by increasing potassium outflow, can inhibit the nervous system in the brain and the expression of GABA-B receptor can accelerate GIRK activation. GABA-B receptors also affect the fundamental activity of K channels [13]. Moreover, GABA-B receptor signaling is involved in the regulation of binge eating [14]. The activity of the GABA-B receptor mitigates binge eating by inducing hyperpolarization with K^+^ outflow. The GABA-B receptor, which can be proposed as a potential treatment target for Alzheimer’s disease (AD), inhibited oxidative stress damage in the neurons of AD model rats by activating the phosphoinositide 3-kinase (PI3K)/protein kinase B (AKT) signaling pathway [15] (Figure 1). BHF177, a positive allosteric modulator of the GABA-B receptor, can activate GABA-B receptors to induce the expression of insulin receptor substrate 1 (IRS-1), PI3K, and anti-apoptotic factors (including Bcl-2 and mTOR); inhibit apoptotic factors, including BCL2 Associated X (Bax), which is known as an apoptosis regulator by releasing cytochrom C in the inner membrane of mitochondria, in hippocampal tissues; and protect against refractory epilepsy (RE) via IRS-1/PI3K [16]. Studying diverse mechanisms involving GABA, especially in the context of neurotransmitters, offers an entry point for understanding the metabolite-mediated signaling cascade in the brain.

## 3. Roles of GABA in Diverse Organ Systems

### 3.1. Liver

GABA exerts different cellular effects and plays an important role in many cells in many organ systems. Impairments in GABA signaling have significant consequences in a range of human physiological processes and diseases. This includes acting as a signaling molecule in the liver. The activation of GABA signaling can protect the liver from D-galactosamine damage by reducing toxic damage in hepatocytes and reducing the proliferation of bile duct cells [17]. GABA not only decreased liver damage caused by a toxic reagent but also reduced liver ischemia and reperfusion injury by modulating the hepatic insulin signaling and gluconeogenesis pathways [18] (Figure 2). GABA improved insulin resistance through glucose transporter 4 (GLUT4) and reduced the glucagon receptor gene expression to inhibit gluconeogenesis [19]. Moreover, pre-emptive treatment with GABA was shown to protect against severe acute liver damage in mice through anti-apoptosis co-ordinating with the STAT3 pathway [20]. GABA supplementation increased the mRNA expression levels of peroxisome proliferator-activated receptor γ (PPARγ) and GABARs and reduced the expression of toll-like receptor 4 (TLR4)/nuclear factor-κB (NF-B) signaling [21], thus protecting the liver from damage.

Among the metabolic pathways, the GABA transporter is associated with lipid metabolism. An intracellular accumulation of DAG due to excessive fatty acid inflow into the liver can result in the activation of protein kinase C (PKC). PKC increases insulin resistance by inhibiting the phosphorylation of IRS-1. An excess of FFA through the activation of inflammatory TLRs can inhibit the phosphorylation of Akt, leading to ceramide synthesis and ceramide accumulation [22]. The accumulation of specific lipids, including DAG and ceramides, impairs hepatic insulin signaling, leading to a pathological increase in hepatic glucose production. The hepatocyte membrane potential regulates serum insulin and insulin sensitivity. Given the role of the membrane potential in regulating media GABA concentrations, GABA import and export are mediated by ion-dependent transporters in the liver. For example, hepatic vagal nerve activity is inhibited by GABA-A receptor stimulation. Hepatocyte depolarization causes hyperinsulinemia and insulin resistance by reducing hepatic vagal afferent nerve (HVAN) activity and increasing GABA outflow from the liver. Interestingly, the GABA release concentration in obese mice is about 180 μmol/mg, while the GABA release concentration in lean mice is about 100 μmol/mg. The different GABA concentration is due to hepatic depolarization in obese mice. The betaine/GABA transporter (BGT-1/GAT2) primarily acts as a GABA reuptake transporter by inducing hepatic hyperpolarization, which limits GABA signaling to the HVAN and promotes metabolic dysfunction. Studies on GABA transporters could provide additional insight into the mechanisms responsible for preserving metabolic health in people with type 2 diabetes [23] (Figure 2).

The GABA shunt classically refers to a tricarboxylic acid (TCA) cycle detour that converts α-ketoglutarate (α-KG) to succinate and concomitantly breaks down a molecule of GABA through GABA-T (Figure 3). However, in the liver, GABA-T mediates GABA synthesis. Hepatic lipids can activate reversed GABA shunt activity in hepatocytes. Increased GABA-T activity, together with the production of GABA, produces α-KG, leading to gluconeogenesis. GABA-T knockdown in obese mice had no effect on body weight. However, it led to decreases in basal serum insulin and glucose concentrations. The inhibition of hepatic GABA production improves insulin sensitivity primarily by increasing the skeletal muscle glucose clearance, which directly affects blood flow. Thus, GABA-T represents a promising target for decreasing hyperinsulinemia and insulin resistance by limiting hepatic GABA production [24], highlighting the importance of understanding hepatic GABA function and the action of GABARs in various disease conditions. We will discuss how GABA metabolism is involved in liver diseases.

### 3.2. Liver Disease: Non-Alcoholic Fatty Liver Disease (NAFLD)

Non-alcoholic fatty liver disease (NAFLD) is currently the most common liver disease worldwide. It can advance to hepatic steatosis, a consequence of lipid acquisition exceeding lipid disposal [25]. Increased glucose can further induce the development of NAFLD by activating transcription factors, including ChREBP and PPAR gamma coactivator 1-b (PPARGC1B), to activate fat production by the liver. The symptoms of NAFLD include deficiencies in lipolysis in the liver (decreased ATGL/CGL-58 activity), deficiencies in triglyceride (TG) export, and an increase in de novo lipogenesis [26]. In simple steatosis, the storage of lipid-droplet-binding TGs is physiologically inactive. However, these lipid complexes are related to hepatocellular damage and apoptosis in non-alcoholic steatohepatitis (NASH). Interestingly, a Western diet challenge was shown to increase triacylglycerol concentrations and induce the mRNA expression levels of macrophage-1 antigen, a cluster of differentiation (CD) 45, CD68, and NF-κB in the liver, promoting the development of hepatic inflammation [27]. Iron overload in the liver is also known to be involved in the pathogenesis of NASH through oxidative stress. High-fat diets (HFDs) were reported to increase iron concentrations in the livers of experimental animals with steatohepatitis [28] (Figure 3). We will focus on the symptoms of NAFLD and how GABA signaling affects NAFLD progression.

### 3.3. Liver Disease Symptom: Modulation of Fatty Acid Oxidation (FAO)

Dysfunction in lipid metabolism can lead to NAFLD. An increase in fatty acid oxidation (FAO) has been suggested as a treatment for NAFLD. FAO is controlled by PPARα and occurs mainly in the mitochondria, providing a source of energy to generate ATP in low glucose conditions. In mammalian cells, the mitochondria, peroxisomes, and cytochromes mediate FAO. The entry of fatty acids into the mitochondria relies on carnitine palmitoyltransferase 1 (CPT1) situated in the outer mitochondrial membrane. However, since mitochondria lack the ability to oxidize very long-chain fatty acids (VLCFAs), these are preferably metabolized through peroxisomal β-oxidation [25]. Increased growth/differentiation factor-15 (GDF-15) in the liver decreased lipid accumulation and NALFD development in obese mice by increasing FAO in the liver [29]. Moreover, for lipid metabolism, many transcription factors and metabolic signaling are involved in liver diseases. Fibroblast growth factor-21 (FGF21) is a circulating hepatokine that beneficially affects carbohydrate and lipid metabolism and is an autocrine factor induced in adipose tissues. It functions in feed-forward loops to regulate the activity of PPAR, a key transcription regulator of fat production [30]. Moreover, the energy sensor AMP-activated protein kinase (AMPK) plays an essential role in the homeostatic regulation of liver lipids. Activated AMPK signaling pathways were shown to increase FAO and reduce lipid synthesis hepatocytes, thereby ameliorating liver steatosis [31] (Figure 3).

### 3.4. Liver Disease Symptom: Activation of Oxidative Stress

Lipid metabolism is linked to oxidative stress. Oxidative stress is defined as an imbalance between oxidants and antioxidants in favor of the former, resulting in an overall increase in the cellular levels of reactive oxygen species (ROS) [32]. Nuclear factor-erythroid-2-related factor 2 (Nrf2), a transcription factor that activates antioxidant response elements (AREs), plays a central role in stimulating the expression of various antioxidant-associated genes in cellular defenses against oxidative stress [33]. This oxidative stress is a key mechanism of hepatocyte injury and disease progression in patients with NASH. Transcription factor Nrf2 plays a central role in stimulating the expression of various antioxidant-related genes in cellular defenses against oxidative stress (Keap1), which directly activates Nrf2 [34]. Inhibiting mitochondrial oxidative damage was shown to prevent metabolic stress-induced NAFLD in mice [35]. Interestingly, GABA inhibits H_2_O_2_ and ROS formation through p65 signaling and regulates Keap1-Nrf2 signaling to repress oxidative stress. Moreover, GABA protected human umbilical vein endothelial cell (HUVEC) injury from H_2_O_2_-driven oxidative damage by inhibiting caspase 3-dependent apoptosis [36]. In addition, GABA has another mechanism to maintain the cellular redox status by modulating glycogen synthase kinase (GSK)-3β and the antioxidant-related Nrf2 nuclear mass ratio. Therefore, GABA may have potential as a pharmacological formulation in the prevention or treatment of cardiovascular diseases associated with oxidative damage [37].

### 3.5. Liver Disease Symptom: Upregulation of Inflammation

The expansion of peripheral adipose deposits provides buffering capacity, which protects the liver from excessive FFA flux that can promote hepatic lipid accumulation. Within hepatocytes, long-chain fatty acids (LCFAs) are esterified with glycerol-3-phosphate (derived from glycolysis) to form mono-acylglycerols, diacylglycerols (DAG), and TG [38].

The production of DAG has been implicated as a cause of hepatic insulin resistance. The conversion of TG to DAG is mediated by adipose triglyceride lipase (ATGL) [39]. Non-esterified hepatic lipids can induce endoplasmic reticulum (ER) stress, leading to the activation of c-Jun N-terminal kinases and NF-κB, two major regulators of inflammatory pathways aggravating hepatic insulin resistance and increasing intrahepatic cytokine production [40]. Deregulated NF-κB activation had a notable effect on the development of hepatic steatosis, insulin resistance, inflammation, fibrosis, and cancer [2]. HFD increased NF-kB activation in mice, which is directly related to chronic inflammation in the liver and fat, hepatic steatosis, and whole-body insulin resistance [41]. Interestingly, GABA regulates inflammation by macrophage cell fate. GABA inhibits interleukin (IL)-1β production by inflammatory macrophages during macrophage maturation. This is due to GABA-dependent mitochondrial protein succinylation, suggesting GABA treatment as a new therapeutic strategy for macrophage-related inflammatory diseases [42]. GABA protected lipopolysaccharide (LPS)-induced inflammation in bovine mammary epithelial cells (MAC-T cell line), by reducing TLR4, NF-κB p65, and MyD88 mRNA expression [43]. However, the physiological significance of the effect of GABA on inflammation remains an open question and the subject of active investigations.

### 3.6. Liver Disease Symptom: Misregulation of ER Stress and Insulin Signaling

The ER is a major site of lipid synthesis in hepatocytes, and ER homeostasis in the liver is important for maintaining membrane lipid composition and controlling both intrahepatic and plasma lipid homeostasis. Since the accumulation of ectopic TG in hepatocytes leads to steatosis, it is important to understand ER-dependent lipid homeostasis in the liver in the first stage of NAFLD. Chronic ER stress, which is another important ER function regulating protein homeostasis, directly affects liver lipid metabolism by inducing de novo lipogenesis, and ER stress is indirectly involved in very-low-density lipoprotein (VLDL) secretion, insulin signaling, and autophagy. Conversely, increasing the hepatocellular lipid content causes chronic ER stress [44]. One of the representative symptoms of ER stress is insulin resistance. An imbalance in lipid contents, such as the activation of TG-levels-induced ER stress, influences insulin resistance [45]. Moreover, ER stress can cause hepatic insulin resistance by increasing de novo lipogenesis and directly interfering with insulin signaling by activating the c-Jun N-terminal kinase (JNK) and IκB kinase (IKK) pathways [46]. ER stress is also regulated by the metabolic signaling pathways. AMPK was shown to enhance insulin sensitivity either by directly regulating PI3K or by suppressing the negative feedback loop of IRS1 regulation by inhibiting mTOR/S6K [47]. AMPK enhances glucose transporter GLUT4 regulation, which is a key regulator of insulin resistance [48], since tumor necrosis factor-α (TNF-α) plays a critical role in the development of NAFLD and progression to NASH by upregulating key molecules associated with lipid metabolism, inflammatory cytokines, and fibrosis in the liver [49]. This TNF-α induction leads to the PP2C-mediated inactivation of AMPK, which increases the fatty acid levels and, potentially, insulin resistance [50]. AMPK also regulates fatty acid synthesis and plays a potential role in hepatic steatosis [51]. Therefore, AMPK may serve as a promising therapeutic target by reducing ER stress [52].

### 3.7. Liver Disease Symptom: Ferroptosis

Ferroptosis is a novel form of programmed cell death caused by iron-dependent lipid peroxidation. Iron overload caused by metabolic dysfunction can aggravate liver damage in NASH patients [53]. Abnormal iron metabolism, lipid peroxidation, and the accumulation of polyunsaturated fatty acid phospholipids trigger ferroptosis, suggesting ferroptosis as a new strategy for the treatment of liver disease. Emerging evidence indicates that ferroptosis plays a critical role in the pathological progression of NAFLD. Hepatocyte ferroptosis precedes cell apoptosis, which, in turn, leads to liver damage, immune-cell infiltration, and inflammation [54]. Ferroptosis is also associated with ER stress, a cellular state accompanied by the accumulation of unfolded or misfolded proteins [55]. The unfolded protein response is also involved in the regulation of DNA-damage-induced ferroptosis. The loss of E3 ubiquitin-protein ligase ring finger protein 113A (RNF113A) was shown to trigger DNA-damage-related ferroptosis [56]. Glutathione peroxidase 4 (GPX4) is known as an inhibitor of ferroptosis, and GPX4 is highly expressed in the liver. GPX4 protects the liver from lipid peroxidation, performing an essential role in liver function and liver cell survival [57]. Recent studies showed that ferroptosis genes are involved in GABA. We previously mentioned that GABA regulates the Keap1-Nrf2 and Notch signaling pathways [36]. Nrf2-related anti-oxidative stress is strongly associated with ferroptosis suppression. Nrf2 silencing dramatically reduced cystine/glutamate transporter (SLC7A11) levels [58]. The SLC7A11/GPX4 pathway functions as a canonical defense against ferroptosis by assisting intracellular glutathione (GSH) synthesis and alleviating lipid peroxidation [59]. Moreover, the GABA-B receptor agonist Taurine reduced ferroptosis and alleviated oxidative stress by regulating the GABA-B/AKT/GSK3/β-catenin signaling pathway. Therefore, clinical studies aimed at activating GABA function may offer new opportunities for treating ferroptosis-mediated liver disease.

### 3.8. Liver Disease Symptom: Induction of DNA Damage

Hepatocytes in NAFLD display the hallmarks of replication stress, including a slow replication fork progression and the activation of an S-phase checkpoint (ATR signaling). Replication-associated DNA lesions accumulate in NAFLD hepatocytes. Patients with NASH reportedly have higher levels of oxidative DNA damage in the liver than patients with other liver diseases [60]. The nucleotide pool imbalance occurring during NAFLD is a key driver of replication stress. Proliferating mouse NAFLD hepatocytes exhibited replication stress with alterations in the replication fork speed and activation of the ATR pathway, which is sufficient for DNA breaks [61]. Lipid overload in proliferating human hepatocytes can lead to replication stress, consequently causing DNA damage. Hepatocyte DNA damage through lipid oxidative stress activates cGAS-STING signaling and leads to the development of sterile inflammation, which drives the pathological process of NAFLD. Injury-induced hepatocyte necrosis or apoptosis can result in the release of nuclear DNA or mtDNA, which can behave as damage-associated molecular proteins (DAMPs) to trigger innate immune responses, giving rise to a sterile inflammation in the liver [62]. Therefore, GABA represents a promising target for decreasing oxidative stress and DNA damage. ROS-driven DNA damage is a by-product of metabolism in hepatocytes, and DNA damage was increased in NAFLD as a consequence of elevated mitochondrial FAO and inadequate mitochondrial respiratory chain activity [63]. Impairment of mitochondria oxidation leads to active fatty acid metabolism [64]. These studies suggest that the regulation of both GABA and fat metabolism is important in liver disease (Figure 3). Future clinical studies aimed at activating GABA function might reveal an attractive therapeutic strategy for DNA-damage-driven NAFLD.

### 3.9. Liver Cancer, GABA, and DNA Damage

Patients with NAFLD are exposed to the risk of HCC. The prevalence of NAFLD-related HCC is rising worldwide [65]. NASH patients have much higher risks of HCC compared to patients with simple hepatic steatosis. HCC is associated not only with NAFLD but also with NAFLD-related metabolic diseases, including obesity and diabetes [66]. For example, PI3K signaling, which regulates metabolism, cell growth, and cell survival, is activated by insulin [67]. PI3K transgenic mice developed steatosis, steatohepatitis, and liver cancer [68]. AKT is downstream of the PI3Ks and is, thus, a major effector of insulin signaling. AKT activation promotes hepatic lipogenesis and modifies the activity of a multitude of downstream targets through phosphorylation. AKT activation led to fatty liver disease and hypertriglyceridemia, whereas AKT inhibition protected against hepatic steatosis [69]. AKT signaling is controlled by phosphatase and tensin homolog (PTEN). PTEN dephosphorylates the lipid second messenger, phosphatidylinositol 3,4,5-trisphosphate (PIP3), a direct product of PI3K. Thus, PTEN is an important negative modulator of the insulin-signaling pathway by antagonizing PI3K–AKT signaling [70]. PTEN loss-of-function mutations, PTEN deletion, or low PTEN expression enhanced insulin sensitivity and promoted hepatic steatosis, steatohepatitis, fibrosis, and liver cancer [71,72,73].

Notably, the combination of oxidative damage and the proliferative response seems to promote carcinogenesis. During early liver cancer development in mouse NAFLD models, oncogene activation led to DNA damage and chromosomal instability [74]. DNA damage due to oxidative stress leads NASH into hepatocellular carcinoma. GABA may perform ROS scavenging and induce Nrf2 activity in liver cancer to reduce oxidative stress. Interestingly, GABA is known to activate PI3K/AKT signaling in the liver. Whereas GABA has beneficial effects on liver biology, GABA-dependent AKT signaling leads to liver cancer. Questions remain regarding the extent to which modulating GABA in the liver can confer a therapeutic benefit. We can suggest that symptoms should be alleviated from NASH to NAFLD through the activity of GABA and FAO before hepatocellular carcinoma deterioration in NASH (Figure 3).

### 3.10. Beta Cells

GABA is a non-proteinogenic amino acid and neurotransmitter that is produced in the islets at levels as high as in the brain. GABA is synthesized by glutamic acid decarboxylase (GAD) [75]. GABA is a product of glutamate. Its metabolism involves the TCA cycle in the pancreatic islets and depends mainly on three enzymes: the synthetic enzyme GAD, the catabolic enzyme GABA transaminase (GABA-T), and succinic semialdehyde dehydrogenase (SSADH). Glucose and glutamine are the most important sources providing glutamate, the substrate for GAD. Subsequently, GAD decomposes glutamic acid to form GABA. GABA is degraded to succinate by GABA-T and SSADH in two steps. The first catabolic step involves the transfer of GABA to α-KG by transamination, which results in the formation of succinic semialdehyde and glutamate. In the second step, succinic semialdehyde is oxidized to succinate with the reduction of nicotinamide adenine dinucleotide (NAD) to NADH. β cells release GABA through a synaptic-like microvesicle (SLMV)-mediated mechanism, which is Ca^2+^-dependent exocytosis, during membrane depolarization [76]. In human β cells, GABA was shown to exert stimulatory effects on proliferation with anti-apoptotic effects [77]. GABA released from islet β cells can act on α and β cells through the paracrine and autocrine pathways. β-cell membrane depolarization through GABA stimulation and VDCC-induced Ca^2+^ influx activated the PI3K/Akt signaling pathway [76]. PI3K/Akt acts as a downstream mediator of the insulin receptor-2 signaling cascade and plays a vital role in protecting β cells from apoptosis while inducing their growth and differentiation [78] (Figure 2).

### 3.11. Adipose Tissues and Skeletal Muscles

Adipose tissues play a key role in the modulation of systemic energy homeostasis in response to physiological stimuli. In obese visceral adipose tissues, increases in the proinflammatory response are positively associated with insulin resistance (Figure 2). Adipose tissue inflammation is a key mediator linking obesity to metabolic complications. GABA was reported to reduce obesity-induced adipose tissue macrophage (ATM) infiltration in subcutaneous inguinal adipose tissue (IAT). GABA also ameliorated systemic insulin resistance in HFD-fed mice and concurrently enhanced the insulin-dependent glucose uptake in IAT [79].

Skeletal muscle is the main metabolic organ consuming ingested glucose in lean individuals with a normal glucose tolerance. Damage to skeletal muscle endothelium following intracellular cascade defects in obese patients led to insulin resistance in this tissue [80]. GABA-A receptor activation modulated skeletal muscle function through muscle sympathetic nerve activity [81] (Figure 2). Further studies on GABA in peripheral tissues may provide an important target for therapeutic intervention for tissue damage.

### 3.12. Clinical Perspective

The various functions of GABA may illuminate new targets for the treatment of metabolic diseases. Since it is important to regulate the action of GABA and GABARs in brain biology, some drugs can inhibit GABA as well as modulate GABARs. With the GABA mechanism, clinical trials are commonly used in brain disease. GABA receptor agonist drugs have been used for anesthesia, anxiety, and insomnia. Abecarnil is a partial agonist, acting selectively at the benzodiazepine site of the GABA-A receptor [82]. Barbiturates are a class of depressant drugs chemically derived from barbituric acid [83]. Muscimol binds to the same site on the GABA-A complex as GABA itself, unlike other GABAergic drugs, such as barbiturates and benzodiazepines, which bind to separate regulatory sites [84]. Propofol is the drug used almost exclusively to induce general anesthesia, and has largely replaced sodium thiopental [85]. Moreover, zolpidem regulates GABA-A, suggesting zolpidem treatment may be a new strategy to control motor symptoms in Parkinson’s disease [86]. GABA receptor antagonists produce stimulant and convulsant effects and are mainly used for counteracting the overdoses of sedative drugs. Clozapine is a tricyclic dibenzodiazepine, classified as an atypical antipsychotic agent [87]. Bicuculline is a phthalide-isoquinoline compound and a light-sensitive competitive antagonist of GABA-A receptors [88]. These clinical shreds of evidence suggest that GABA-A receptor agonists and antagonists may be used in different metabolic disorders including liver disease. As with GABA-A receptors, there are drugs that identify a GABA-B receptor agonist. Lesogaberan (AZD3355) was developed for the treatment of gastroesophageal reflux disease (GERD) [89]. Recently, AZD3355 has been used for the treatment of NASH as a GABA-B agonist [90]. A compelling future direction for the field of GABA biology and metabolism will be the repositioning of GABA target drugs for metabolic disorders. While these clinical trials were developed by the mechanism of GABA in physiology, studies to validate the clinical trial drugs for their repositioning in different cell systems and in vitro for another mechanism would be important for understanding the GABA function. This challenge suggests we should bridge the gaps between the in vitro, animal, and clinical studies of GABA. 

## 4. GABA Metabolism

GABA biosynthesis is tightly linked to nitrogen metabolism (Figure 4). Glutaminase can catalyze the hydrolysis of the amide group of glutamine to form glutamate and ammonia [91]. Glutamine synthase (GS) catalyzes the reverse reaction, an ATP-dependent reaction responsible for the formation of glutamine from glutamic acid and ammonia. GS is a key enzyme in the glutamic acid–glutamine cycle. It plays a major role in the brain homeostasis of glutamic acid, glutamine, and ammonia. GABA is synthesized by the decarboxylation of glutamic acid by GAD [92]. Glutamate required for GABA synthesis is synthesized from two pathways: glutamine and TCA cycle-derived α-KG. Glutamate dehydrogenase (GDH) catalyzes the reaction between glutamate, α-KG, and ammonia using NAD+ or NADP+ as coenzymes [93]. GDH is important in glutamic acid and GABA neurotransmission because it directly regulates glutamic acid concentrations and indirectly regulates GABA levels by changing the availability of precursors. GDH is potentially inhibited by GTP and activated by ADP [94]. GABA decomposition requires the conversion of GABA-T to GABA to succinate semialdehyde (SSA) through an amino group transition with glutamic acid and α-KG, both of which are auxiliary substrates. The glutamine–glutamate/GABA circuit transfers glutamine from astrocytes to neurons and transfers neurotransmitter glutamic acid or GABA from neurons to astrocytes. Much more glutamine is transferred from astrocytes to glutamate than from GABAergic neurons [95]. Glutamate is a metabolic precursor of glyceraldehyde 3-phosphate through glyceraldehyde 3-phosphate circuits. GABA synthesis is unique among neurotransmitters. Homocarnosine and pyrrolidinone are two distinct isoenzymes of GAD with major effects on GABA metabolism in the human brain [5]. Therefore, the modulation of the GABA biosynthetic and metabolic pathways may offer new opportunities for GABA-linked metabolic disease therapy.

## 5. Conclusions and Future Perspectives

In this review, we summarized the diverse roles of GABA in several organs, and its original role as a neurotransmitter. Increasing evidence suggests that GABA has a marked effect on metabolism at the cellular level of several organs, including the pancreas, muscle, fat, and liver, in addition to its role as an inhibitory neurotransmitter, with a major role in regulating various alert states. Current studies showed that GABA works through intercellular actions, such as autocrine and paracrine functions, in peripheral organs. However, the extent of modulating the absolute levels of GABA in the blood and each metabolic organ needed to confer a therapeutic benefit in metabolic disorders remains unclear. The cellular function would be different in many different organs and physiological conditions as well as in vitro systems; therefore, further studies are necessary to define the different GABA effects in in vitro and in vivo systems to understand the limitations of GABA research.

Many studies have shown that the biosynthesis of GABA is relevant to important metabolites, including α-KG and glutamate. GABA-T enzymes are generally known to convert α-KG to glutamate and decompose GABA to succinic semialdehyde. However, GABA-T can synthesize GABA in the liver. Lipid accumulation in the liver can induce insulin resistance by activating GABA-T to activate gluconeogenesis through the accumulation of α-KG. The modulation of these key TCA metabolites is important not only for energy metabolism but also for obesity-related diseases. Therefore, whether clinical interventions with GABA transporters and GABA-T might mitigate disease progression, including obesity and insulin resistance, remain to be investigated. Currently, the cellular effects of glutamate and glutamine are well-known. However, the role of intracellular GABA has not been elucidated. Further studies on the intracellular effects of GABA in the liver and other metabolic diseases, as well as the cellular compartmentalization of GABA and other related nitrogen metabolites in liver diseases, are necessary. This approach may provide an opportunity to move beyond the role of GABA as a neurotransmitter.

## Figures and Tables

**Figure 1 ijms-24-04584-f001:**
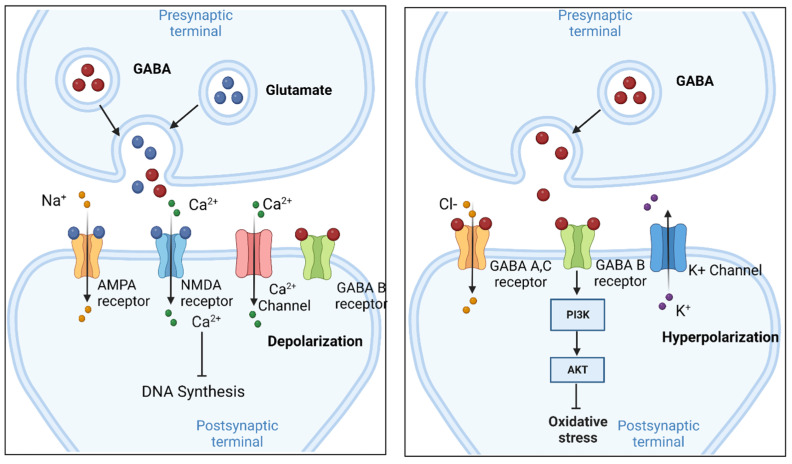
The role of GABA and glutamate as neurotransmitters. In the early stages of cortical neurogenesis, glutamate and GABA depolarize cells in the ventricular region of the mouse embryo neocortex (left panel). Glutamate binds its receptor, which is known as N-methyl-D-aspartate receptor (NMDA receptor or NMDAR), to depolarize the neuron. Glutamate selectively interacts with α-amino-3-hydroxy-5-methyl-4-isoxazole propionic acid (AMPA) and GABA can bind to gamma amino butyric acid (GABA)-B receptors. GABA and glutamate increase the concentration of Ca^2+^ in ventricular zone cells, and decrease DNA synthesis by activating voltage-gated Ca^2+^ channels. Although most metabolites regulate depolarization of the synapse, GABA-B involves hyperpolarization (right panel). GABA-B receptor activates potassium outflow channels through G-protein signaling to induce hyperpolarization. GABA-B receptor activates phosphoinositide 3-kinase (PI3K)/protein kinase B (AKT) signaling and anti-apoptotic factors to relieve oxidative stress.

**Figure 2 ijms-24-04584-f002:**
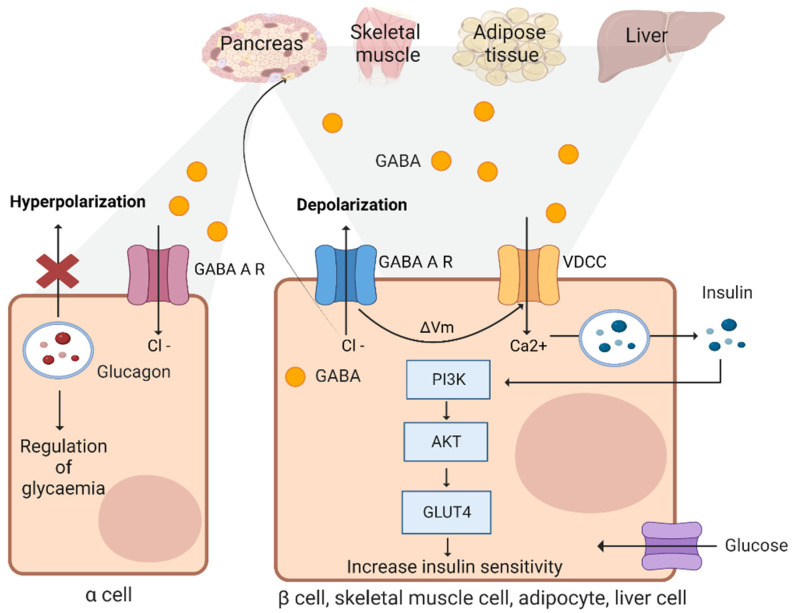
The function of GABA in various organs. Schematic representation of gamma-aminobutyric acid (GABA) on transmission in the pancreas, skeletal muscle, adipose tissue, and liver. GABA induces hyperpolarization in alpha cells in the Islets of Langerhans in the pancreas by increasing the inflow of Cl^−^. The membrane hyperpolarization inhibits the action of glucagon secretion. GABA regulates glycemia by inhibiting alpha-cell function. On the other hand, GABA induces depolarization in beta cells in the Islets of Langerhans in the pancreas, skeletal muscle, adipocyte, and liver cells, by increasing the inflow of Ca^2+^. In this case, GABA activates the PI3K-AKT pathway to increase insulin sensitivity by upregulating glucose transporter 4 (GLUT4) expression.

**Figure 3 ijms-24-04584-f003:**
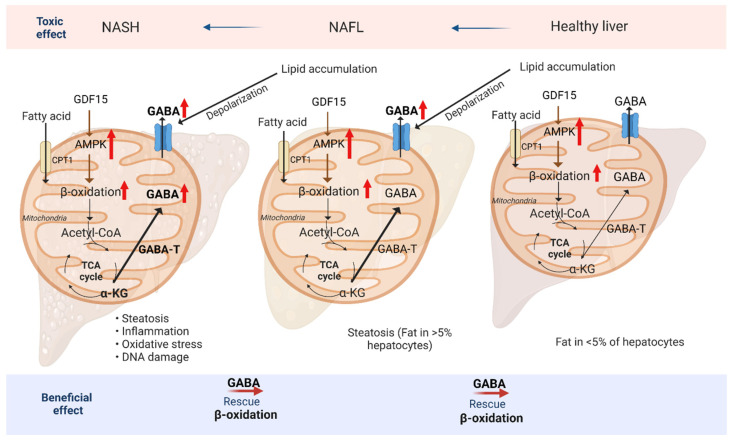
GABA and fatty acid metabolism reduce the symptoms of liver disease. Schematic representation of dysfunction of lipid metabolism, which is linked to non-alcoholic fatty liver disease (NAFLD, NAFL). Growth differentiation factor 15 (GDF15) activates 5′-AMP-activated protein linase (AMPK) pathway to increase fatty acid oxidation (FAO, b-oxidation), which directly affects the accumulation of fatty acid in non-alcoholic steatohepatitis (NASH). Activation of FAO decreases oxidative stress and DNA damage in NASH and NAFL to a healthy liver. Moreover, GABA transaminase (GABA-T) converts alpha-ketoglutarate (a-KG) to gamma-aminobutyric acid (GABA), suggesting GABA directly regulates membrane potential in liver. Lipid accumulation in liver disease induces hepatocyte membrane depolarization and increases GABA outflow, which leads to hyperinsulinemia and aggravates the symptoms of NASH and NAFL.

**Figure 4 ijms-24-04584-f004:**
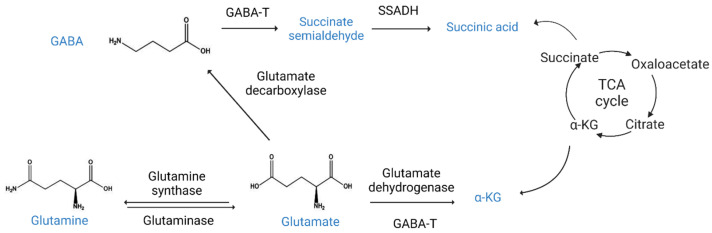
GABA biosynthesis and catabolism. The major biosynthesis pathway of gamma-aminobutyric acid (GABA) from glutamine and alpha-ketoglutarate (a-KG). Glutaminase catalyzes the reaction of transforming glutamine to glutamate. The reverse reaction is controlled by glutamine synthase. Glutamate dehydrogenase (GDH) transforms glutamate to a-KG. Glutamate decarboxylase (GAD) catalyzes the reaction of transforming glutamate to GABA. GABA transaminase converts GABA to succinate semialdehyde. Succinic semialdehyde dehydrogenase (SSADH) catalyzes the reaction of transforming succinate semialdehyde (SSA) to succinic acid.

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
