# Peer review of "Gamma-Aminobutyric Acid Signaling in Damage Response, Metabolism, and Disease"

_ijms, 2023, doi:10.3390/ijms24054584_

Round 1
Reviewer 1 Report
1. The English need improvement since there are some grammatical and syntax errors in the manuscript. For example,
· in line number 78, the word “synapse” may be as “the synapse”;
· in line number 115, “STAT3” as “the STAT3”;
· in line number 121, “pancreas” as “the pancreas”;
· in line number 171, “cluster” as “a cluster”;
· in line number 184, “liver” as “the liver”;
· in line number 272, “fatty” as “the fatty”;
· in line number 306, “to DNA” as “for DNA”;
· in line number 328, “a downstream” as “downstream”.
The grammar mistakes which are not mentioned hecre are also to be checked and corrected properly.
2. There are some typing mistakes as well, and authors are advised to carefully proof-read the text. For example,
· in line number 64, the word “isoxazolepropionic” may be as “isoxazole propionic”;
· in line number 125, “cell,” as “cells,”;
· in line number 222, “status-by” as “status by”;
· in line number 258, “Conversely,increasing” as “Conversely, increasing”;
· in line number 317, “lead to” as “leads to”;
· in line number 407, “synptoms” as “symptoms”.
The typos not mentioned here are also to be checked and corrected properly.
3. Check the abbreviations throughout the manuscript and introduce the abbreviation when the full word appears the first time in the abstract and the remaining for the text and then use only the abbreviation (For example, triglyceride (TG), endoplasmic reticulum (ER), long-chain fatty acids (LCFAs), etc.,). Make a word abbreviated in the article that is repeated at least three times in the text, not all words to be abbreviated.
4. The literature search should be described in detail. The authors are encouraged to include the database, search engines (like PubMed, ScienceDirect, Google scholar etc.,), the keywords used etc., which may be included since it is a review article.
5. The authors may include or address the following for better outcome of the manuscript. How many articles obtained from each of the search engines? What is the inclusion and exclusion criteria? What is the type of article included in this manuscript? How many articles are included for this manuscript? A diagram depicted the literature search should be included for better understanding and outcome.
6. The figure legends should be improved and a proper footnote should be given. All legends should have enough description for a reader to understand the figures without having to refer back to the main text of the manuscript. For example, the necessary abbreviations should be given in the figure 2 (GABA, GLUT-4).
Author Response
Response to Reviewer 1 Comments
We really appreciate your all comments. We have added and edited our manuscript to address your terrific comments. The details are as below and we also attached a word version for the figures.
Point 1: The English need improvement since there are some grammatical and syntax errors in the manuscript. For example,
- in line number 78, the word “synapse” may be as “the synapse”;
- in line number 115, “STAT3” as “the STAT3”;
- in line number 121, “pancreas” as “the pancreas”;
- in line number 171, “cluster” as “a cluster”;
- in line number 184, “liver” as “the liver”;
- in line number 272, “fatty” as “the fatty”;
- in line number 306, “to DNA” as “for DNA”;
- in line number 328, “a downstream” as “downstream”.
Response 1: We really appreciate your nice corrections on this manuscript. We have all updated errors based on your comments.
Point 2: There are some typing mistakes as well, and authors are advised to carefully proof-read the text. For example,
- in line number 64, the word “isoxazolepropionic” may be as “isoxazole propionic”;
- in line number 125, “cell,” as “cells,”;
- in line number 222, “status-by” as “status by”;
- in line number 258, “Conversely,increasing” as “Conversely, increasing”;
- in line number 317, “lead to” as “leads to”;
- in line number 407, “synptoms” as “symptoms”.
The typos not mentioned here are also to be checked and corrected properly.
Response 2: Thank you so much again for these corrections. We have edited all mistakes.
Point 3: Check the abbreviations throughout the manuscript and introduce the abbreviation when the full word appears the first time in the abstract and the remaining for the text and then use only the abbreviation (For example, triglyceride (TG), endoplasmic reticulum (ER), long-chain fatty acids (LCFAs), etc.,). Make a word abbreviated in the article that is repeated at least three times in the text, not all words to be abbreviated.
Response 3: Thank you for your great comments. We have fixed this error.
Point 4: The literature search should be described in detail. The authors are encouraged to include the database, search engines (like PubMed, ScienceDirect, Google scholar etc.,), the keywords used etc., which may be included since it is a review article.
Response 4: Thank you, we have added the PubMed ID in the reference.
Point 5: The authors may include or address the following for better outcome of the manuscript. How many articles obtained from each of the search engines? What is the inclusion and exclusion criteria? What is the type of article included in this manuscript? How many articles are included for this manuscript? A diagram depicted the literature search should be included for better understanding and outcome.
Response 5: To address reviewer’s comment, we added literature flow diagram about searching results to included studies. We follow the searching strategy on search features including main keywords and screen recent 5 years paper with our topics using National Center for Biotechnology Information PubMed and google engines. A criteria for paper selection, the type of articles and number of papers are describe in Figure 1 in this rebuttal.

Figure 1. Literature flow diagram: Searching results to included studies.
Point 6: The figure legends should be improved and a proper footnote should be given. All legends should have enough description for a reader to understand the figures without having to refer back to the main text of the manuscript. For example, the necessary abbreviations should be given in the figure 2 (GABA, GLUT-4).
Response 6: We appreciate your great comments. We have updated full names of every abbreviation in each figure.
Also, we added important information in each figure, as described below:
In Figure 1; Glutamate binds its receptor, which is known as N-methyl-D-aspartate receptor (NMDA receptor or NMDAR), to depolarize the neuron.
In Figure 3; Moreover, GABA transaminase (GABA-T) converts alpha-ketoglutarate (a-KG) to gamma-aminobutyric acid (GABA).

Reviewer 2 Report
I consider this simple review article, relevant to the topic, well written and a contribution to updating knowledge in this area and alerts to the need for further research to clarify GABA mechanisms in metabolic diseases.
I only suggest that in the summary they should mention, for example, in paragraph 14 …. damage and its beneficial and toxic effects on the progression of the disease
I would have liked to have developed more the issue of binge eating, which as we know can contribute to the development of obesity. This has become a major health concern is associated with the development of various diseases such as type 2 diabetes, cardiovascular disease, cancer and eating disorders and studies suggest that gamma-aminobutyric acid (GABA) receptor signaling type B (GABA B R ) is involved in the regulation of binge eating and the influence of a diet high in saturated fat on binge eating.
Author Response
Response to Reviewer 2 Comments
We really appreciate your all comments. We have added and edited our manuscript to address your terrific comments.
Point 1: I only suggest that in the summary they should mention, for example, in paragraph 14 …. damage and its beneficial and toxic effects on the progression of the disease
Response 1: Thank you so much for this amazing comment. We have added this point in the abstract; with this review, we suggest that further research is necessary to develop GABA's beneficial and toxic effects on metabolic disease progression.
Point 2: I would have liked to have developed more the issue of binge eating, which as we know can contribute to the development of obesity. This has become a major health concern is associated with the development of various diseases such as type 2 diabetes, cardiovascular disease, cancer and eating disorders and studies suggest that gamma-aminobutyric acid (GABA) receptor signaling type B (GABA B R ) is involved in the regulation of binge eating and the influence of a diet high in saturated fat on binge eating.
Response 2: I really appreciate your terrific suggestion on adding the topic of GABA and appetite. Since this review is more focused on liver physiology and metabolism, we will cover this amazing topic in a future review article.
Reviewer 3 Report
In this review, the authors focus on the role of gamma-aminobutyric acid (GABA) in the liver, adipose tissue, and skeletal muscle. There are detailed descriptions and discussions of the effects of GABA on liver-related diseases, helping readers to understand GABA metabolism, physiological pathways, biosynthesis, and cellular functions that may differ from those in the brain. Some of the studies cited are conducted in vitro, while others are not. So in Section 5 it would be worthwhile to briefly discuss the limitations of in vitro and in vivo findings and how the two types of investigations might complement each other in the future to further our knowledge about GABA. How to bridge the gaps between in vitro, animal, and clinical studies will continue to be a challenge. Section 3.12 does not offer a particular point of view, which should be shaped by the authors' research experiences and understanding of GABA. It reads like a simple descriptive summary.
Author Response
Response to Reviewer 3 Comments
We really appreciate your all comments. We have added and edited our manuscript to address your terrific comments.
Point 1: In this review, the authors focus on the role of gamma-aminobutyric acid (GABA) in the liver, adipose tissue, and skeletal muscle. There are detailed descriptions and discussions of the effects of GABA on liver-related diseases, helping readers to understand GABA metabolism, physiological pathways, biosynthesis, and cellular functions that may differ from those in the brain. Some of the studies cited are conducted in vitro, while others are not. So in Section 5 it would be worthwhile to briefly discuss the limitations of in vitro and in vivo findings and how the two types of investigations might complement each other in the future to further our knowledge about GABA. How to bridge the gaps between in vitro, animal, and clinical studies will continue to be a challenge. Section 3.12 does not offer a particular point of view, which should be shaped by the authors' research experiences and understanding of GABA. It reads like a simple descriptive summary.
Response 1: We really appreciate your wonderful comment. We have added the limitation of in vitro and in vivo system in discussion (Section 5) as below.
The cellular function would be different in many different organs and physiological conditions as well as in vitro systems, therefore, further studies are necessary to define the different GABA effects in vitro and in vivo systems to understand the limitations of GABA research.
Moreover, to address the questions on how to bridge the gaps between in vitro, animal, and clinical studies will continue to be a challenge, we suggest the repositioning of clinical trial drugs in Section 3.12. Clinical perspective, as following.
While these clinical trials were developed by the mechanism of GABA in physiology, the studies to validate the clinical trial drugs for repositioning in different cell systems and in vitro for another mechanism would be important to understand the GABA function.
Especially, this Section 3.12 is introducing the clinical trials of GABA, we have added connecting sentence to show our idea.
Reviewer 4 Report
In the manuscript, the authors summarized the diverse roles of GABA in several organs, and its original role as a neurotransmitter. Although the work fits the scope and the goals of the International Journal of Molecular Sciences, the work must undergo minor revision before further evaluation.
The major drawbacks are as follows:
Cellular function of gamma-aminobutytic acid in the brain section. Line 47. What exactly is a signaling molecule and how does it work?
Line 58. What is or what does GABAergic mean?
At the bottom of figure 1, it is not necessary to put the name of GABA as well as AMPA, since it has already been defined previously.
Lines from 87 to 95. The authors could make a figure of what they want to explain in these lines, it would be better illustrated and it would be better understood. I consider it relevant to the work and would facilitate the reader's understanding of the idea.
Lines from 89 to 91. Can you go into more detail and say what K channels mean and imply? I think this part is not clear.
Line 98. What is Bax? not defined
Roles of GABA in diverse organ systems section. Line137. Could the authors give some more information regarding those levels of concentration of which they speak?
Liver disease: non-alcoholic fatty liver disease (NAFLD) section. Lines from 172 to 173. The authors told: "Iron overload in the liver is also known to be involved in the pathogenesis of NASH through oxidative stress". To what extent and how does iron affect this? Is there data to confirm it? How does iron work here? The authors confirm that high-fat diets increase iron concentrations, so what diet would be appropriate for a person who has chronic or fairly severe anemia?
Liver disease symptom: modulation of fatty acid oxidation section. Line 203. Protein kinase controls glucose and lipid metabolism. It is a protein that adds phosphates to other molecules such as sugars or other proteins. This causes other molecules to become active or inactive. Next, I find it interesting that the authors further explain the role that protein kinase plays.
Liver disease symptom: ferroptosis section. What role does or could GABA play in ferroptosis and how would it play?
Adipose tissues and skeletal muscles section. Lines from 377 to 378. Some reference should be added.
Resultados de traducción
Resultado de traducc
Author Response
Response to Reviewer 4 Comments
We really appreciate your all comments. We have added and edited our manuscript to address your terrific comments. The details are as below and we also attached a word version for the figures.
Point 1: Cellular function of gamma-aminobutytic acid in the brain section. Line 47. What exactly is a signaling molecule and how does it work?
Response 1: Thank you so much for your comment. We have added the sentence as GABA is an inhibitory neurotransmitter by regulating ionic channel inflow and outflow.
GABA is known as an inhibitory neurotransmitter by regulating ionic channel inflow and outflow. For example, GABA A,C Receptor are ligand-gated ion channel. When GABA act as a neurotransmitter at GABA A,C receptor, Cl- channel opens and induces Cl- inflow to occur hyperpolarization. Also, GABA B receptor is G-protein coupled receptor for GABA. GABA B receptor coupled to potassium channel when GABA act as a neurotransmitter potassium channel opened to induce potassium outflow to occur hyperpolarization.
Point 2: Line 58. What is or what does GABAergic mean?
Response 2: We appreciate your point. To address your comment, you have added the meaning of GABAerging, which is affecting and producing GABA. For example, GABAergic neuron means that neuron produces GABA, GABAergic drug includes GABA agonist, antagonist, modulators, reuptake inhibitors and enzymes.
Point 3: At the bottom of figure 1, it is not necessary to put the name of GABA as well as AMPA, since it has already been defined previously.
Response 3: Thank you so much for your suggestion. We don’t need to add AMPA and GABA altogether in Figure 1, but other reviewer requested more details in this figure. Thanks, again.
Point 4: Lines from 87 to 95. The authors could make a figure of what they want to explain in these lines, it would be better illustrated and it would be better understood. I consider it relevant to the work and would facilitate the reader's understanding of the idea.
Response 4: To address your great comment, we have added the more detailed mechanism in GABARs in brin function as below.
GABA-A is required to trigger the Ca2+ response. For example, GABA was shown to activate ionic GABARs to stimulate cerebrovascular angiogenesis and promote neurovascular binding in cerebrovascular endothelial cells [12]. The mechanism of GABA is that GABARs-dependent Ca2+ response leads to promote cerebrovascular angiogenesis and induce neurovascular binding for activating brain function.
Point 5: Lines from 89 to 91. Can you go into more detail and say what K channels mean and imply? I think this part is not clear.
Response 5: To explain more details, we have added that GIRK Channel activation induce hyperpolarization by increasing potassium outflow.
Point 6: Line 98. What is Bax? not defined
Response 6: To understand more clearly, we have explain more about BCL2 Associated X (Bax), which known as an apoptosis regulator by releasing cytochrom C in the inner membrane of mitochondria.
Point 7: Roles of GABA in diverse organ systems section. Line137. Could the authors give some more information regarding those levels of concentration of which they speak?
Response 7: To make clear our idea, we have added the different GABA concentration from this research paper (reference 23, Figure 2 in the rebuttal draft). The GABA release concentration in obese mice is about 180 μmol/mg, while the GABA release concentration in lean mice is about 100 μmol/mg. Different GABA concentration is due to hepatic depolariztion in obese mice. Moreover, inducing Hepatocyte membrane hyperpolarization decreases GABA release in obese mice liver slice by expressing K+ channel, Kir2.1 (Figure 2F).

Figure 2. The different GABA release in between obese and lean mice.
Point 8: Liver disease: non-alcoholic fatty liver disease (NAFLD) section. Lines from 172 to 173. The authors told: "Iron overload in the liver is also known to be involved in the pathogenesis of NASH through oxidative stress". To what extent and how does iron affect this? Is there data to confirm it? How does iron work here? The authors confirm that high-fat diets increase iron concentrations, so what diet would be appropriate for a person who has chronic or fairly severe anemia?
Response 8: Thank you so much for your great point. We have added iron overload in the liver is also known to be involved in the pathogenesis of NASH through oxidative stress (with new reference 28). Ferrous ion catalyzes the Fenton reaction, in which hydrogen peroxide is efficiently converted to the reactive oxygen species hydroxyl radical, the presence of excess iron in tissues increases oxidative stress and harms normal biological functioning, possibly leading to various diseases including chronic liver diseases. It seems to give a high-fat diet challenge in a person who has chronic or severe anemia would be an interesting strategy to increase iron concentrations, however, we should be more careful to coordinate the complex physiological condition in metabolic disorders.
Point 9: Liver disease symptom: modulation of fatty acid oxidation section. Line 203. Protein kinase controls glucose and lipid metabolism. It is a protein that adds phosphates to other molecules such as sugars or other proteins. This causes other molecules to become active or inactive. Next, I find it interesting that the authors further explain the role that protein kinase plays.
Response 9: Thank you so much for your comments. We defined the different enzymatic functions of AMPK and PHD3 in acetyl-CoA carboxylase (ACC) for fatty acid oxidation (FAO) in muscle function. AMPK phosphorylates ACC1 and 2, which converts acetyl-CoA into malonyl-CoA. Phosphorylation of ACC1 and ACC2 inhibits their enzymatic activity by changing structure of ACC. The ACC phosphorylation site is serine 222 residue. ACC2, which is associated with the outer mitochondrial membrane, generates malonyl-CoA to inhibit carnitine palmitoyl transferase 1 (CPT1), which mediates the first step of long-chain fatty acid transport into the mitochondria. By blocking CPT activity, ACC2 prevents substrate entry into mitochondrial FAO. Thus, by phosphorylating both ACC isoforms, AMPK coordinates the rates of cellular fat synthesis and catabolism. In this review, we focus dynamic regulation of metabolism in liver, which related to GABA metabolism. FAO is not directly affecting GABA metabolism, suggesting GABA may coordinate TCA cycle with FAO by regulating a-KG levels.
Point 10: Liver disease symptom: ferroptosis section. What role does or could GABA play in ferroptosis and how would it play?
Response 10: We described that GABA regulates Keap1-Nrf2 and Notch signaling pathways. Nrf2-related anti-oxidative stress is strongly associated with ferroptosis suppression. Nrf2 silencing dramatically reduced cystine/glutamate transporter (SLC7A11) levels [58]. SLC7A11/GPX4 pathway functions as a canonical defense against ferroptosis by assisting intracellular glutathione (GSH) synthesis and alleviating lipid peroxidation.
Moreover, we have added a clinical bridge that GABA B receptor agonist Taurine reduced ferroptosis and alleviated oxidative stress by regulating the GABA B/AKT/GSK3/β-catenin signaling pathway.
Point 11: Adipose tissues and skeletal muscles section. Lines from 377 to 378. Some reference should be added.
Response 11: Thank you so much for your great suggestion. We focus on the GABA effect on liver in this review, we will mention more details on GABA in adipose tissue and skeletal muscle in future reviews. We will add more research next time.
